# Modulated Electro-Hyperthermic (mEHT) Treatment in the Therapy of Inoperable Pancreatic Cancer Patients—A Single-Center Case-Control Study

**DOI:** 10.3390/diseases9040081

**Published:** 2021-11-03

**Authors:** Flora Greta Petenyi, Tamas Garay, Dorottya Muhl, Blanka Izso, Adam Karaszi, Erika Borbenyi, Magdolna Herold, Zoltan Herold, Attila Marcell Szasz, Magdolna Dank

**Affiliations:** 1Faculty of Information Technology and Bionics, Pazmany Peter Catholic University, 1083 Budapest, Hungary; petenyigreta@gmail.com (F.G.P.); garay.tamas@itk.ppke.hu (T.G.); zsozsoblanka@gmail.com (B.I.); 2Division of Oncology, Department of Internal Medicine and Oncology, Semmelweis University, 1083 Budapest, Hungary; muhl.dorottya@med.semmelweis-univ.hu (D.M.); adam.karaszi@gmail.com (A.K.); borbenyi.erika@med.semmelweis-univ.hu (E.B.); herold.zoltan@med.semmelweis-univ.hu (Z.H.); szasz.attila_marcell@med.semmelweis-univ.hu (A.M.S.); 3Department of Internal Medicine and Hematology, Semmelweis University, 1088 Budapest, Hungary; herold.magdolna@med.semmelweis-univ.hu

**Keywords:** hyperthermia, concomitant, pancreatic neoplasms, modulated electro-hyperthermia

## Abstract

Our present oncological treatment arsenal has limited treatment options for pancreatic ductal adenocarcinoma (PDAC). Extended reviews have shown the benefits of hyperthermia for PDAC, supporting the perspectives with the improvements of the treatment possibilities. METHODS: A retrospective single-center case-control study was conducted with the inclusion of 78 inoperable PDAC patients. Age-, sex-, chemotherapy-, stage-, and ascites formation-matched patients were assigned to two equal groups based on the application of modulated electro-hyperthermia (mEHT). The EHY2030 mEHT device was used. RESULTS: A trend in favor of mEHT was found in overall survival (*p* = 0.1420). To further evaluate the potential beneficial effects of mEHT, the presence of distant metastasis or ascites in the patients’ oncological history was investigated. Of note, mEHT treatment had a favorable effect on patients’ overall survival in metastatic disease (*p* = 0.0154), while less abdominal fluid responded to the mEHT treatment in a more efficient way (*p* ≤ 0.0138). CONCLUSION: mEHT treatment was associated with improved overall survival in PDAC in our single-center retrospective case-control study. The outcome measures encourage us to design a randomized prospective clinical study to further confirm the efficiency of mEHT in this patient cohort.

## 1. Introduction

The present oncological treatment arsenal has limited treatment options for pancreatic cancer, despite all efforts and multiple studies [1]. Most of this difficult and aggressively lethal disease is pancreatic ductal adenocarcinoma (PDAC, 90%), with a five-year survival rate of less than 10% and its median survival is only 4–6 months in advanced stages [2,3,4,5]. The symptoms and the clinical shreds of evidence of this complex disease often appear late during tumor progression. Many manifestations of this disease are refractory against chemo- and radiotherapy treatments and possess an immunosuppressive microenvironment [6]. The epidemiologic observations show a pessimistic future: pancreatic cancer will be the second leading cause of cancer-related deaths in the United States by 2030 [7]. Current treatment options for PDAC are that only the early stage is operable, and most of the treatment is chemotherapy in all stages of the disease [8]. When the disease is locally advanced, chemotherapy has low impact on quality of life (QoL), but with metastatic disease chemotherapy and disease progression can significantly worsen the patient’s QoL [9]. Various chemotherapies differently modify the patients’ QoL, showing a spectrum of tolerability and a careful indication based on the condition of the patient [10]. Higher physical and mental QoL scores are associated with longer survival and can predict a better prognosis [11].

The most common treatments are chemo-combinations including the FOLFIRINOX protocol or the combination of Gemcitabine with platinum agents or nab-paclitaxel [12]. Unfortunately, no such effective medication exists for PDAC other than for other solid malignancies like breast or colon cancer, and the QoL is also worse than in many of the treatments in other localizations. The PDAC is presently resistant to most of the traditional treatments and only a limited number of early-stage patients are completely operable [13]. Consequently, finding new, effective therapies would be mandatory. The challenge to increase the treatments’ efficacy is the exceptional tumor-heterogeneity of PDAC, which can be captured both at a molecular and cellular level and the substantial interaction with the host tissues, including the dense neostroma and the immune system as well. The task needs a multidisciplinary approach and expertise [14], as the key to optimal treatment is to overcome the structural and functional barriers of immune reactions [15]. The treatment strategy should be formed by a team of multidisciplinary experts [16], providing the availability to a broad spectrum of diagnostic and therapeutic facilities [17]. 

One of these multidisciplinary methods, the complementary application of hyperthermia, combines the conventional therapies with advantageous biophysical effects against the tumor cells. A phase II clinical trial, which started in 2017 (HEATPACK [18]) and was recently published [19], demonstrated the feasibility of the hyperthermia application and the achievement of better survival and patient QoL results. Extended reviews show the benefits of the hyperthermia for PDAC [20,21], supporting the perspectives with the improvements of the treatment possibilities. The challenge, which also adds to the complexity, is to simultaneously consider the enormous structural heterogeneity of the tumor-tissues and the bioelectrical, thermal, and density-inhomogeneity interactions of cancerous lesions.

Hyperthermia methods can be divided into full-body and local heating methods [22]. One of the latest advancements in local oncologic hyperthermia methods is called modulated electro-hyperthermia (mEHT), developed for heterogenic energy absorption, and treating usually advanced cases when the conventional therapies became ineffective. Technically, mEHT is a precision capacitive coupled impedance matched method [23]. It works by radiofrequency (RF) of 13.56 MHz frequency, producing RF current through the target [24]. The main challenge of the extreme heterogeneity of PDAC fits the active mechanism of mEHT, which targets the tumor cells selectively. The functional and bioelectric features of the malignant cells drive the selective process, which, together with the dynamic interactions and thermal exchanges, kills the cancer cells in an apoptotic way. Without going into details, mEHT exploits the biophysical differences of cancer cells that their energy absorption on the membrane rafts is different than those of healthy host cells, furthermore, damage-associated molecular patterns (DAMPS) will occur and all of these eventually lead to programmed or immunogenic tumor cell death [25,26]. The mEHT method re-sensitizes the refractory cases, can act as monotherapy as preclinical results and case studies suggested [25,27,28,29], and increases the survival time and QoL in parallel. Another line of complexity is the ability to systematically eliminate the micro and macrometastases, which are unfortunately very frequent even at the time of diagnosis. The immune system would be the most evident support for this task, and this effect had been shown by hyperthermia as early as 1986 [30]. The mEHT method produces immunogenic cell-death, and through this, forms tumor-specific immune effects [31,32]. Numerous clinical studies were performed with mEHT on the topic of pancreatic cancer [33,34,35,36].

The aim of the present retrospective case-control study was to assess the additional benefit of mEHT to conventional systemic therapy of PDAC and provide the basis for a prospective clinical study to confirm the concomitant utilization of mEHT.

## 2. Materials and Methods

The study was conducted in accordance with the WMA Declaration of Helsinki; handling of patient data was in accordance with the General Data Protection Regulation issued by the European Union. The study was approved by the Regional and Institutional Committee of Science and Research Ethics, Semmelweis University (SE_IKEB_8_2017).

### 2.1. Study Design and Patient Selection

Here, we present a retrospective single-center case-control study including 78 stage-III and -IV pancreatic cancer patients not eligible for surgery, treated at the Division of Oncology, Department of Internal Medicine and Oncology, Semmelweis University, Budapest, Hungary, during the period between September 2016 and November 2019. All pancreatic cancer diagnosis were set during routine diagnostic protocol by histological examination between 26 December 2014, and 17 October 2019. Data collection was terminated on 30 June 2021. Patients were assigned into two groups based on the addition of mEHT treatment, having an equal number of participants (39 patients/group).

### 2.2. Treatment Protocols and Matching of Cases and Controls

All patients were treated with standard-of-care chemotherapy regimens, which were decided at the tumor board sessions individually, based on European and national guidelines. The active study group and the control group were matched in all subcategories of the treatment. At first, each casepatient was individually matched to a control-patient by age (±5 years), sex, and chemotherapy administered during mEHT treatment. To reach a higher similarity in the case and control patients’ overall status, the presence or absence of distal metastases and the emergence of ascites in the patients’ history were considered as well as a matching criterion by generating case-control pairs. No cirrhosis or any other co-morbidities causing ascites were present in any of the cases. The RECIST guideline v1.1 was used to evaluate response to treatment [37].

### 2.3. Hyperthermia Treatment

The EHY2030 device (Oncotherm Ltd., Budaörs, Hungary) was used to administer the complementary mEHT treatment. Case patients underwent at least 21 mEHT treatment (Table 1). Each session lasted between 30–60 min, with a step-up power output between 60–150 W, until reaching the patients’ maximum tolerability. No special adverse event due to conducting mEHT was observed throughout the study.

### 2.4. Statistical Analysis

Statistical analysis was performed with R for Windows 4.1.0 (R Core Team, 2021, Vienna, Austria). Overall- (OS) and progression-free (PFS) survival of patients was defined as the length of time from the date of PDAC diagnosis until the death from any cause and until disease progression or death from any cause, respectively. Follow-up of patients was terminated on 30 June 2021. Patients alive at this time point were right-censored. Hazard ratios (HR) and its 95% confidence intervals (95%CI) were calculated using the Cox proportional hazards regression model and survival rates with Kaplan–Meier analyses supported by log-rank tests. Prior comparison, the normality of continuous data was assessed by Quantile-Quantile plots and normality was rejected both for age and survival data. Continuous and categorical variables between case and control group were compared using the Wilcoxon–Mann–Whitney rank-sum test and N-1 chi-squared test, respectively. The N-1 chi-squared test was calculated using the method described previously [38]. The Cochran–Mantel–Haenszel test and linear model were used for subgroup analysis. Results were expressed as mean ± standard deviation, mean ± standard deviation (median (range)) and as the number of observations (percentage) for parametric and non-parametric continuous, and categorical data, respectively. Two-sided *p*-values < 0.05 were considered as significant.

## 3. Results

A total of 78 patients, 39 matched case-control pairs were included in the study. Baseline characteristics of patients are summarized in Table 2. Patients were further divided into sub-cohorts based on the presence of metastasis, ascites and metastasis-ascites combined. A total of 20 case + 24 control and 19 case + 15 control study participants were assigned to the with and without metastasis groups; 13 case + 18 control and 26 case + 21 control to the with and without ascites groups; and 13 case + 10 control, 6 case + 5 control, 8 case + 16 control and 12 case + 8 control to the none, only ascites, only metastasis and both ascites and metastasis groups, respectively (Appendix A).

Comparison of cases and controls revealed that both the median OS (*p* = 0.0301) and PFS (0.0258) of mEHT treated patients was significantly higher. One-year OS (*p* = 0.0240) and one-year PFS (*p* = 0.0455) was higher in the case group, but two- and three-year survival of patients did not differ (Table 1). Similar results were found in subgroup analyses in the case of OS and PFS. In addition, OS and PFS calculated from the first mEHT treatment was lower in patients with ascites and/or metastasis, lower OS after the last mEHT treatment was found in patients with ascites, while PFS did not differ in any of the subgroups (Appendix A).

### 3.1. Analysis of Overall- and Progression-Free Survival

Although there was a significant difference between the OS of the two groups based on group-comparisons, results of survival analysis revealed that the two groups did not differ from each other (*p* = 0.1420). After reviewing the individual survival curves, the following was observed. A practically constant difference could have been observed between the two survival curves until the second year, where 91.6% of patient deaths occurred, from which those patients with longer survival times were approximately the same (Figure 1). Age of study participants (*p* = 0.7430) or the location of the tumor (pancreatic head vs. body/tail, *p* = 0.2920) had no significant effect on OS, when investigating all the 78 study participants together. Analyzing the data within the two groups individually, age did not affected OS (Control group: *p* = 0.1980; mEHT group: *p* = 0.6670). Location of the tumor (head vs. body vs. tail of pancreas) had a marginal effect on patient survival in controls (*p* = 0.0728), while no difference was found in the mEHT group (*p* = 0.9880). However, if the location was grouped as body/tail or head of pancreas, the adjuvant mEHT treatment was more beneficial where the tumor was located in the body/tail of the pancreas (Cox HR: 0.3493, 95% confidence interval (95%CI): 0.1546–0.7890, *p* = 0.0114; Figure 2).

To further evaluate the potential beneficial effects of mEHT treatment, emerging distant metastasis or ascites in the patients’ history were investigated whether the metastatic disease stage or ascites influence the mEHT treatment. In our observation, neither PDAC patients with metastatic disease nor without metastasis had statistically different OS if treated with mEHT (with metastasis HR: 0.6786, *p* = 0.2160; without metastasis HR: 0.8768, *p* = 0.7370). It must be mentioned that the shape of survival curves in controls was similar to that shown on Figure 1, but in the mEHT group the metastatic and non-metastatic curves were very similar (Appendix A). When we considered the presence of ascites, we have observed that the patients possessing no abdominal fluid responded to the mEHT treatment in a more efficient way (HR advantage of mEHT: 0.5248, 95%CI: 0.2757–0.9989, *p* = 0.0496; Figure 3 and Appendix A).

PFS was analyzed in a similar manner as OS. Progressive disease was observed in 28 and 26 case and control patients, while sudden death in 11 and 13 patients, respectively (benefit of mEHT, HR: 0.6173, 95%CI: 0.3914–0.9736, *p* = 0.0380; Figure 4). Metastatic patients of the mEHT groups had worse survival (Cox HR: 2.1690, 95%CI: 1.0780–4.3640, *p* = 0.0300), while no difference was found in the control group (*p* = 0.1960; Figure 5). The presence of ascites, location of the tumor and age did not affect PFS in any of the study groups.

### 3.2. Analysis of Survival after the First and Last mEHT Treatments

Survival times after the first and last mEHT treatment were also evaluated. Age (first mEHT: *p* = 0.9890; last mEHT: *p* = 0.7050), the time elapsed between the diagnosis of PDAC and the first mEHT treatment (first mEHT: *p* = 0.9060; last mEHT: *p* = 0.7560), and the location of the tumor (first mEHT: *p* = 0.9550; last mEHT: *p* = 0.7060; Appendix A) had no effect on any of the mEHT-connected survival times. As expected, the higher the applied number of mEHT treatments/patient was, better ‘survival-after-first-treatment’ was found (HR: 0.9747, 95%CI: 0.9574–0.9922, *p* = 0.0048), however, no such effect could have been observed for the ‘survival-after-last-treatment’ (*p* = 0.2900). Both the emergence of metastases (HR: 2.4830, 95%CI: 1.1900–5.1820, *p* = 0.0154) and ascites (HR: 2.4925, 95%CI: 1.2650–4.9090, *p* = 0.0083) had a significant effect on ‘survival-after-first-treatment’. After the last treatment survival times were significantly affected by only the presence of ascites (HR: 2.3849, 95%CI: 1.1940–4.7620, *p* = 0.0138; Figure 6 and Figure 7).

Until the end of mEHT treatment, progression was observed in 22 of the 39 case patients (56.41%) with an average 5.70 ± 3.18 months ‘time to progression’ after the first mEHT treatment. The remaining patients of the mEHT group had stable disease at the end of treatment; neither partial nor complete response was observed, but progression was observed later for all of these patients (4.71 ± 4.63 months after the completion of mEHT). Progression to both the end of the study and the end of mEHT treatment was also examined. Only the presence of metastasis affected significantly the PFS of patients (progression to the end of the study, HR: 2.4165, 95%CI: 1.2070–4.8360, *p* = 0.0127; progression to the end of mEHT treatment, HR: 2.0960, 95%CI: 0.8745–5.0240, *p* = 0.0971, Figure 8).

## 4. Discussion

In oncology, hyperthermia is the artificial heating of the tumor and its surrounding to fever-like temperature via different biophysical techniques [22], which ultimately has tumoricidal effects on cancer cells [22,39,40,41,42]. Superficial- and deep-seated tumors have been treated with hyperthermia since the late 1970s and 1980s, respectively [25,43,44]. Two major methods have been developed: whole-body and regional hyperthermia [22] and one of the latest advancements in regional hyperthermia is mEHT [45]. Early studies investigating the effect of hyperthermia in pancreatic cancer have included mainly whole-body hyperthermia, while latest investigations, in most cases, apply only regional hyperthermia, including mEHT. Previous studies concluded, that progression-free- and overall survival of patients improved if complementary hyperthermia was adjusted [29,30,33,34,35,43,46,47,48,49,50,51,52,53,54,55,56,57,58,59,60]. Furthermore, higher maximum output power of the hyperthermia device has been positively correlated with partial response rate [54], a possible correlation between the time from diagnosis to first mEHT treatment and the survival time from first mEHT treatment have been proposed [33], and the combination of hyperthermia with various herbal remedies [29,59,60] further improved patient response to oncotherapy.

In ductal adenocarcinoma of the pancreas, the oncological guidelines are developing, but options are still limited and the prognosis remains dismal [61]. To facilitate the efficiency of currently available systemic oncotherapies [62], we undertook an approach to improve outcome of pancreatic cancer patients with the utilization of mEHT. Our hypothesis was to extend the overall survival of the treated “case” patient cohort. The positive effect of mEHT on better disease control rate, progression-free and overall survival is known [33,35]. Based on available literature and previous clinical investigations, a logical presumption was to suggest facilitating effect of mEHT on traditional chemotherapy and radiotherapy [33,34,35,36]. 

Here, we investigated the possible benefit from adding mEHT to standard treatment of pancreatic adenocarcinoma patients and have observed both significant differences and trends in overall survival benefiting mEHT treated patients in case-control pairs matched by age, sex and chemotherapy. In our further subset analysis, PDAC patients with the metastatic disease did not differ from those patients without metastasis, while in the control group a significant difference in PFS was found, and a similar trend in the survival curve of OS. The observations may suggest that metastatic PDAC patients may be one that group that can benefit the most from the mEHT treatment. When the presence of ascites was subjected to analysis, we have observed that the patients possessing less abdominal fluid responded to the mEHT treatment in an improved matter. However no difference in the times-to-progression was observed. 

The effect of the location of the tumor on OS and PFS was also investigated. It was found that if the tumor is located in the body/tail of the pancreas, the addition of mEHT has a significant benefit on overall survival of PDAC patients. Pancreatic body/tail tumors are known to be more aggressive as those are usually diagnosed in later stages than the tumors of the pancreatic head [3,63]. A Chinese study investigating the combined effect of mEHT with supplementary traditional herbal remedy therapy on metastatic pancreas cancer patients with ascites have found similar result to ours, better absorption of ascites, response to treatment and QoL was observed compared to those patients who received conventional chemotherapy and regular drainage [29].

To our knowledge, only one study [35] investigated the survival of patients after the first/last mEHT treatments in various settings. Dani et al. [35] reported that patients with metastases have worse survival than those of without, furthermore, they have reported that the number of mEHT therapies has no significant effect on patient survival. The above observation were also confirmed in the present study and we could have expanded the knowledge on mEHT-related survival times with the following. The emergence of ascites is a poor sign of patient survival, even with the utilization of mEHT. Furthermore, we have observed a negligible effect of age, location of the tumor within the pancreas and the elapsed time between the diagnosis of PDAC and the first mEHT treatment on mEHT-related patient survival.

To summarize our results, we conclude that inoperable PDAC patients can benefit from mEHT treatment. Patients with no ascites formation responds to the therapy most and metastatic patients treated with mEHT showed significantly slower progression formation.

## 5. Conclusions

A retrospective single-center case-control study was conducted to assess the additional benefit of mEHT to conventional systemic therapy of PDAC. All the observations found are in line with previous findings, the addition of mEHT as a complementary treatment in inoperable PDAC improves overall survival, significantly better results can be achieved in ascitic and metastatic cases. Limitations of our work is the relatively small sample size, and the retrospective design. The results of the study encourage us to design a randomized prospective clinical study to further confirm the efficiency of mEHT in this patient cohort and further reinforces those observations that hyperthermia should be considered as a routine treatment option in oncology care.

## Figures and Tables

**Figure 1 diseases-09-00081-f001:**
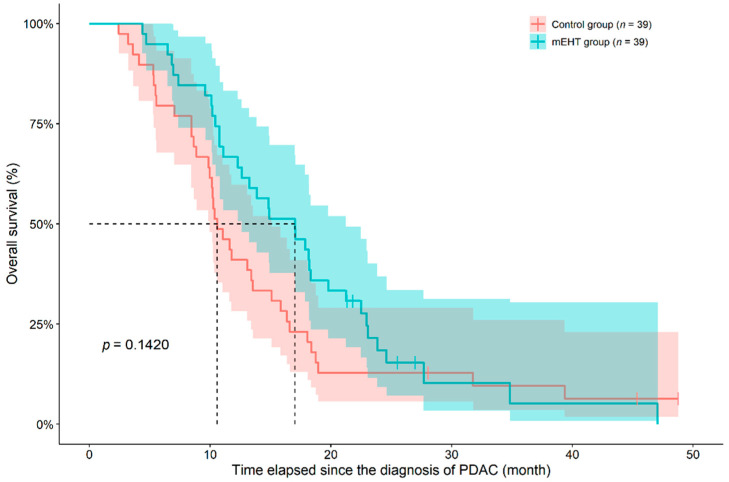
Overall survival of inoperable pancreatic ductal adenocarcinoma (PDAC) patients in case-control pairs matched for age (±5 years), sex, and chemotherapy receiving during modulated electro-hyperthermia (mEHT) treatment. The dotted line and the lighter colored intervals represent median survival and the asymmetrical 95% confidence interval, respectively. The *p*-value of log-rank test was drawn.

**Figure 2 diseases-09-00081-f002:**
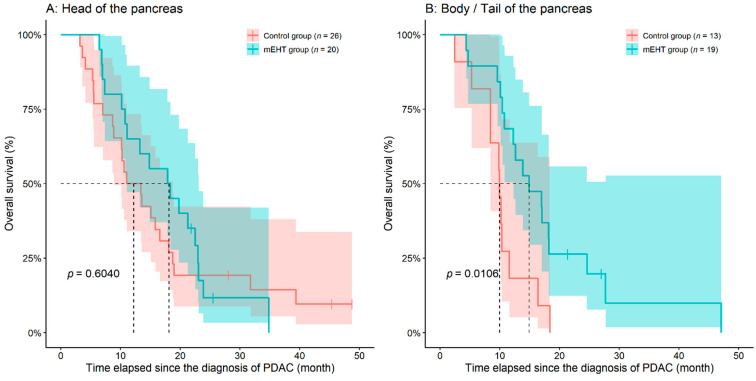
Modulated electro-hyperthermia (mEHT) treatment has a significant beneficial effect if the tumor is located in the body/tail of the pancreas (**B**). If the tumor is in the head of the pancreas (**A**), a partly similar trend can be observed. The dotted line and the lighter colored intervals represent median survival and the asymmetrical 95% confidence interval, respectively. *p*-value of log-rank test was drawn. PDAC: pancreatic ductal adenocarcinoma.

**Figure 3 diseases-09-00081-f003:**
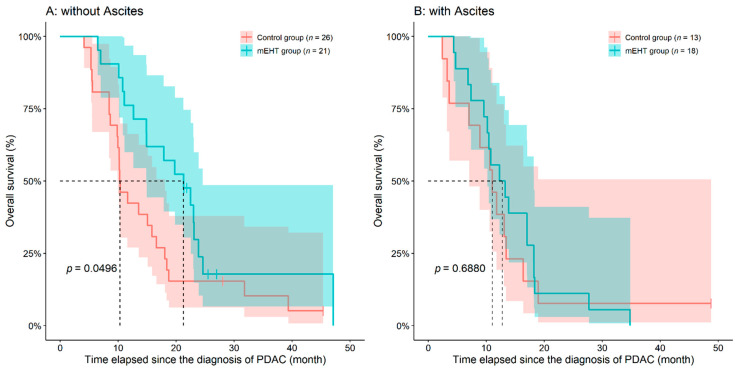
Inoperable pancreatic ductal adenocarcinoma (PDAC) patients treated with adjuvant modulated electro-hyperthermia (mEHT) had significantly better survival if no ascites was present (**A**). However, with the emergence of ascites (**B**) only a slight trend could have been observed in favor of mEHT treatment. The dotted line and the lighter colored intervals represent median survival and the asymmetrical 95% confidence interval, respectively.

**Figure 4 diseases-09-00081-f004:**
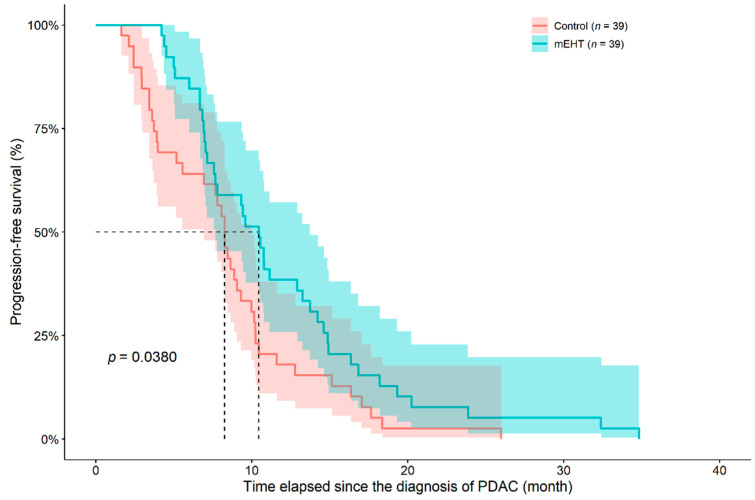
Progression-free survival of inoperable pancreatic ductal adenocarcinoma (PDAC) patients in case-control pairs matched for age (± 5 years), sex, and chemotherapy received during modulated electro-hyperthermia (mEHT) treatment. The dotted line and the lighter colored intervals represent median survival and the asymmetrical 95% confidence interval, respectively.

**Figure 5 diseases-09-00081-f005:**
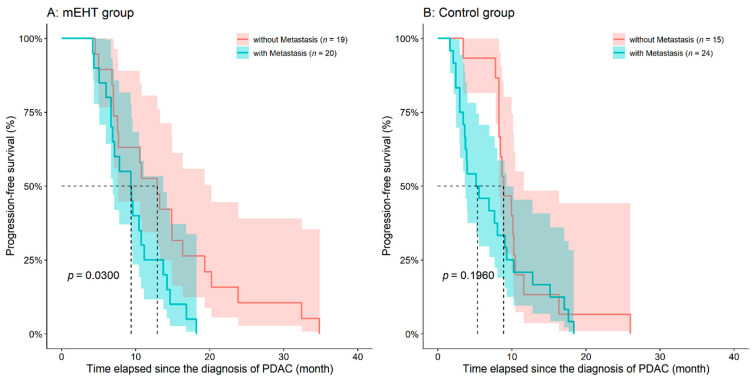
Differences in progression-free survival of inoperable pancreatic ductal adenocarcinoma (PDAC) patients who have been treated with (**A**) or without (**B**) adjuvant modulated electro-hyperthermia (mEHT) with or without the presence of metastases. The dotted line and the lighter colored intervals represent median survival and the asymmetrical 95% confidence interval, respectively.

**Figure 6 diseases-09-00081-f006:**
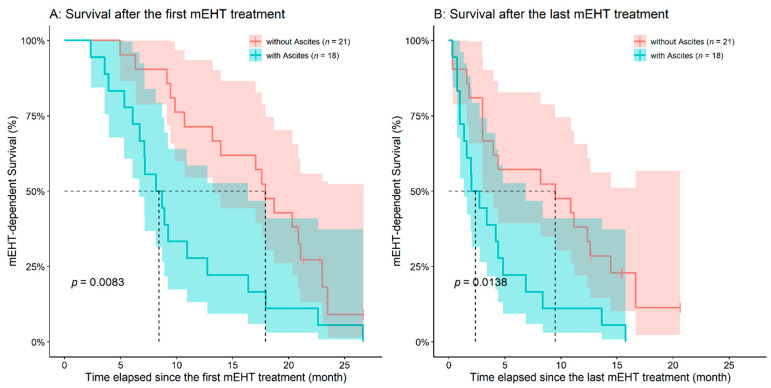
The emergence of ascites had a negative effect on the survival times after both the first (**A**) and the last (**B**) modulated electro-hyperthermia (mEHT) treatment. The dotted line and the lighter colored intervals represent median survival and the asymmetrical 95% confidence interval, respectively.

**Figure 7 diseases-09-00081-f007:**
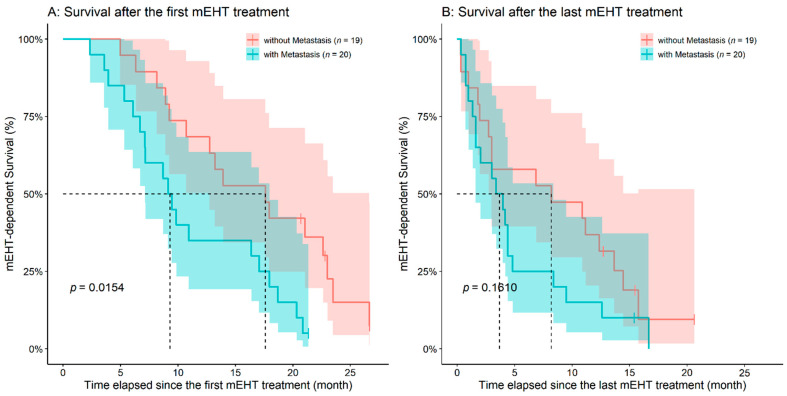
The emergence of metastases had a negative effect on the survival times after the first (**A**) modulated electro-hyperthermia (mEHT) treatment, while only a trend can be observed in the case of survival times after the last mEHT treatment (**B**). The dotted line and the lighter colored intervals represent median survival and the asymmetrical 95% confidence interval, respectively.

**Figure 8 diseases-09-00081-f008:**
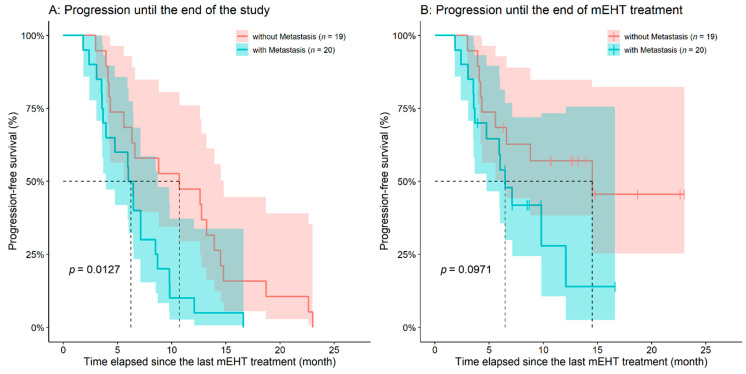
The emergence of metastases had a negative effect on progression-free survival both if the progression was investigated until (**A**) the end of the study and (**B**) until the end of the mEHT treatment. The dotted line and the lighter colored intervals represent median survival and the asymmetrical 95% confidence interval, respectively.

**Table 1 diseases-09-00081-t001:** Data of modulated electro-hyperthermia (mEHT) treatment applied to the case group.

Parameter	Value
Number of mEHT sessions	59.18 ± 26.22 (64 (21–154))
Elapsed time from diagnosis to mEHT treatment (days from pathological diagnosis)	99.56 ± 147.88 (46 (2–718))
Applied applicator (cm)	30
Average duration of sessions (min)	59.5
Frequency of treatments/patients (/week)	2–3
Protocol (fit to patient’s tolerability)	Step-up
Power (W)	From 60 to 150

**Table 2 diseases-09-00081-t002:** Basic demographic and clinical characteristics are summarized of the case and control groups (mean ± SD (median and range)). Unit of frequency data is the number of observations (percentage).

Parameter	mEHT Treated (*n* = 39)	Control (*n* = 39)	*p*-Value
Age (years)	65.90 ± 9.90 (67 (45–84))	66.02 ± 8.73 (67 (45–78))	0.8927 ^1^
Male gender	18 (46.2)	18 (46.2)	matched
Location of the tumor - Head of the pancreas - Body of the pancreas - Tail of the pancreas	20 (51.3) 14 (35.9) 5 (12.8)	26 (66.7) 8 (20.5) 5 (12.8)	0.3030 ^2^
Chemotherapy protocol: - GEM monotherapy - FOLFORINOX - GEM + cisplatin - GEM + 5-fluorouracil + leucovorin - GEM + oxaliplatin	24 (61.5) 8 (20.5) 5 (12.8) 1 (2.6) 1 (2.6)	24 (61.5) 8 (20.5) 5 (12.8) 1 (2.6) 1 (2.6)	matched matched matched matched matched
Without radiologically detected ascites	21 (53.8)	26 (66.7)	0.2504 ^2^
Without distant metastasis	19 (48.7)	15 (38.4)	0.3642 ^2^
Overall survival (month)	16.96 ± 8.72 (17.02 (4.4–47.1))	14.19 ± 10.86 (10.58 (2.4–48.8))	0.0301 ^1^
Progression-free survival (month)	11.87 ± 7.05 (10.45 (4.2–34.8))	8.53 ± 5.37 (8.25 (1.6–26.0))	0.0258 ^1^
One-year OS	26 (66.7)	16 (41.0)	0.0240 ^2^
Two-year OS ^#^	6 (15.4)	5 (12.8)	0.6761 ^2^
Three-year OS ^#^	1 (2.6)	3 (7.7)	0.3481 ^2^
One-year PFS	15 (38.5)	7 (17.9)	0.0455 ^2^
Two-year PFS	1 (2.6)	2 (5.1)	0.5585 ^2^
OS after the first mEHT treatment (month)	13.69 ± 7.11 (12.75 (2.3–26.7))	–	–
OS after the last mEHT treatment (month)	6.57 ± 5.83 (4.17 (0.3–20.6))	–	–
PFS after the first mEHT treatment (month)	8.60 ± 5.45 (6.60 (1.8–23.0))	–	–
PFS after the last mEHT treatment (month) *	1.48 ± 4.88 (0.72 (−9.0–15.8))	–	–

^1^ Wilcoxon–Mann–Whitney rank sum test, ^2^ N-1 chi-squared test. ^#^ 2 and 4 right-censored case patients had lower observation time than 24 and 36 months, respectively, and one censored control patient had a censor time between 24–36 months. * Negative values in ‘PFS after the last mEHT treatment’ means that progression developed during the mEHT treatments. mEHT: modulated electro-hyperthermia, GEM: Gemcitabine; FOLFORINOX: leucovorin (FOL) + 5-fluorouracil (F) + irinotecan (IRIN) + oxaliplatin (OX), OS: overall survival, PFS: progression-free survival.

## Data Availability

The datasets used and/or analyzed during the current study are available from the corresponding author on reasonable request.

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
