# Peer review of "Modulated Electro-Hyperthermic (mEHT) Treatment in the Therapy of Inoperable Pancreatic Cancer Patients—A Single-Center Case-Control Study"

_diseases, 2021, doi:10.3390/diseases9040081_

Round 1

Reviewer 1 Report

The study is interesting and promising but is poorly reported. The main concerns are inappropriate statistics and multiple inconsistencies in survival analysis. After fixing these errors, the article can be recommended for publishing. There are also a lot of smaller consideration, part of which is mentioned below. It is hard to correct the article line by line, so the authors need make an effort to put the article in order. The statistics must be fully recalculated.

Author Response

Budapest, September 3, 2021

Dear Reviewer,

On behalf of my fellow authors, first of all, I would like to thank you for your attention and opinion on our original article entitled “Modulated electro-hyperthermic (mEHT) treatment in the therapy of inoperable pancreatic cancer patients - a single center case-control study”.  We thank our first Reviewer their careful work. Here we provide answers to the questions our first Reviewer raised.

The study is interesting and promising but is poorly reported. The main concerns are inappropriate statistics and multiple inconsistencies in survival analysis. After fixing these errors, the article can be recommended for publishing. There are also a lot of smaller consideration, part of which is mentioned below.

Thank you very much for your kind criticism, we agree with it, and in the light of that, we have refined our manuscript.

  1. It seems that the authors erroneously chosen the statistical methods. “Comparisons of other parameters between case and control group were investigated using Wilcoxon-Mann-Whitney rank sum test or Fisher’s exact test for continuous and categorical variables, respectively.
  • Why MW? This is a non-parametric test. Do the authors by default consider their data not normally distributed? Why? Contrary, data is considered normal by default, while abnormality of distribution have to be proven, not vice versa. I don’t see the normality test results. The sample size is not small. MW can give substantial bias in parametric data and does not provide standard deviation. This is inappropriate test for normal data.

Normality of continuous data was assessed by Quantile-Quantile plots and the reason behind using Mann-Whitney tests was that neither the distribution of age, nor any of the survival data followed normal distribution (please see the attached figure below). Statistical methods had been updated with the following sentence: “Prior comparison, the normality of continuous data was assessed by Quantile-Quantile plots and normality was rejected both for age and survival data.

  • The same refers to Fisher’s Exact test (FET). It is recommended if the group size is <20 or any subgroup is <5. In this case, n=39 and minimum subgroup n=5, so the Pearson’s chi-square test with N-1 correction (Richardson test) is a proper choice, since FET underestimates the significance. For example (Table 1).

Thank you for the suggestion, comparison of categorical data was updated with N-1 chi-squared tests using the method described by Busing et al. (2016, doi: 10.1002/sim.6808, reference added).

  • Results were expressed as median (range) and as the number of observations (percentage).” The standard results presentation for parametric data is means with standard deviations (errors). The presentation by medians and ranges worsens the comparability of results and prevents the inclusion in meta-analysis. Medians can be reported as an additional robust statistic but not the main one.

Thank you for your suggestion. As some of the data did not follow normal distribution, we choose to present both mean ± SD and median (range) for the non-parametric data. Methods were updated as follows. “Results were expressed as mean ± standard deviation, mean ± standard deviation (median (range)) and as the number of observations (percentage) for parametric and non-parametric continuous, and categorical data, respectively.”

  1. Suboptimal reporting of the results and inconsistencies in survival analysis
  • The mEHT treated patients had longer median overall survival than those of control cases (Table 1, Figure 1).” There is a discrepancy. The wording suggests a significant difference, which is confirmed by reference to Table 1 (p=0.03), while Figure 1 and preceding text suggest that the difference is not significant (p=0.14).

Thank you for bringing our attention to this issue. Results had been completely rewritten as follows. “Although there was a significant difference between the OS of the two groups based on group-comparisons, results of survival analysis revealed that the two groups did not differ from each other (p = 0.1420). After reviewing the individual survival curves, the following was observed. A practically constant difference could have been observed between the two survival curves until the second year, where 91.6% of patient deaths occurred, from which those patients with longer survival times were approximately the same (Figure 1).

  • In our observation, PDAC patients with metastatic disease benefited the most from the treatment and displayed a longer median survival times than the control patients (Control group: 10.5 months; mEHT group: 14.8 months; Figure A1”. There is no Figure 1A, so that the reference is incorrect. In Figure 2A, the difference is about 12 vs 18 months. The indicated difference of 10.5 vs 14.8 months fits Figure 2B, but Figure 2 applies to head and tail-located tumors, not to metastatic disease state, so irrelevant to the text. Moreover, it is not clear why a modest difference in MST of 4.3 month in a smaller sample (N=19+13=32) resulted in p=0.0106 (Figure 2B), while a larger difference of about 7 months in larger sample (N=78) resulted in p=0.1420 only (Figure 1).

As per ‘Author guidelines’ of the journal, items included within the manuscript’s Appendix have to be named as Figure A1, A2, Table A1, A2, etc. The other Reviewer of our manuscript encouraged us to extend the results with additional material. Due to the additionally requested figures and tables, we had decided to introduce a Supplementary file instead of using the Appendix option of the Journal. Furthermore, the whole Results section had been revised, with which amendments the manuscript should be more clear and easier to follow.

  1. Other considerations
  • “All pancreatic cancer diagnosis was set” seems to be “Every (each) pancreatic cancer diagnosis was set” or “All pancreatic cancer diagnoses were set”
  • “Data collection was closed on January 31, 2020.“ “Closed” usually means a prospective trial. For retrospective trial, a more correct expression is “Data collection was terminated on January 31, 2020.“
  • “Patients alive at this time point were” right-“censored.”
  • References to figures. If part A of figure 1 is referred to, it is usually Figure 1A, not Figure A1.

The manuscript had been addressed as suggested. Grammar, typos and a few previous discrepancies have been corrected throughout the text.

  • Table 1 The endpoint name “Overall survival” is inaccurate: this is the “Median overall survival,” and conventionally this is the “Median Survival Time” (MST).

Definition of overall survival was updated in methods.

Yours sincerely,

                                            Prof. Dr. Magdolna Dank

                                             Semmelweis University

Reviewer 2 Report

The manuscript presents a clinical study with the aim to evaluate potential benefits of modulated electro-hyperthermic treatment (mEHT) in the therapy of patients suffering from inoperable pancreatic ductal adenocarcinoma (PDAC). PDAC is a particularly aggressive disease which is usually diagnosed at the late stage when treatment options are limited. Pancreatic adenocarcinoma has a dismal prognosis with less than 10% of patients surviving at 5 years from diagnosis. Therefore, the search for additional treatments which could improve survival of patients and efficacy of standard therapies, such as chemotherapy, is very important. The authors of this study involving 78 PDAC patients, divided in 39 matched case-controls with several chemotherapy protocols, report improved survival in mEHT treated group. The article also investigates the effect of other parameters (e.g. metastasis occurrence, ascites presence) on the outcome of mEHT treatment. Although the presented results are an important contribution towards the body of evidence on mEHT usefulness in therapy of PDAC patients, the article needs careful rewriting and better organization of results and figures: findings need to be presented in a clearer way.  The authors should also provide some other outcome measures for assessing mEHT efficiency (e.g. progression-free survival, tumor response).

Questions and remarks:

  • The text needs careful editing, it contains sentences which are difficult to understand: e.g lines 50-52, lines 150-151, 246-247, 256-257 etc
  • What was the staging of patients?
  • What are the mechanisms behinds selectivity of mEHT toward cancer cells? – please add into Introduction
  • Please provide reference in the Introduction to “mEHT …acts as monotherapy” (line87)

  • The results section is not presented in a clear way, it would help to summarize all the results in a table where the medians of overall survival (and other measures) would be given for the whole groups and subgroups (patients with metastasis, ascites etc) as well. Please, summarize also the influence of all the other factors you assessed (e.g. location of tumor, No. of mEHT sessions etc) on the overall survival and mEHT result. In the present form, it is difficult to follow all the findings.
  • The graphs should be grouped into figure panels in the order in which they are presented step-by-step in the Results text.
  • Can you provide other measures for assessing the impact of mEHT? For example: data on partial response, stable disease, progression free disease or time to progression?
  • Fig A1 and the relevant text (lines 173-179): You report “that patients with metastatic disease benefited most from mEHT” – could you please explain this statement? The shift in OS is about 4 months in the metastasis group, in case of patients without ascites it is about 11 months when mEHT was administered. Please explain what HR of 2.11 and 1.6019 (line 178) refer to. What was the portion of patients having ascites in the groups with and without metastasis? Could it affect the results?
  • Fig A2: there are no dotted lines and colored intervals. In the figure, A is Control group, B is mEHT group, in the figure legend A is “with” and B is “without” mEHT

  • Conclusion at the end of discussion (line 263-265) and sentence (270-271 “best results can be achieved in metastatic cases”) should be rephrased

Author Response

Budapest, September 3, 2021

Dear Reviewer,

On behalf of my fellow authors, first of all, I would like to thank you for your attention and opinion on our original article entitled “Modulated electro-hyperthermic (mEHT) treatment in the therapy of inoperable pancreatic cancer patients - a single center case-control study Thank you to our second Reviewer for the positive feedback on our article. Our answers for the critical comment and the statistical question are below.

The manuscript presents a clinical study with the aim to evaluate potential benefits of modulated electro-hyperthermic treatment (mEHT) in the therapy of patients suffering from inoperable pancreatic ductal adenocarcinoma (PDAC). PDAC is a particularly aggressive disease which is usually diagnosed at the late stage when treatment options are limited. Pancreatic adenocarcinoma has a dismal prognosis with less than 10% of patients surviving at 5 years from diagnosis. Therefore, the search for additional treatments which could improve survival of patients and efficacy of standard therapies, such as chemotherapy, is very important. The authors of this study involving 78 PDAC patients, divided in 39 matched case-controls with several chemotherapy protocols, report improved survival in mEHT treated group. The article also investigates the effect of other parameters (e.g. metastasis occurrence, ascites presence) on the outcome of mEHT treatment. Although the presented results are an important contribution towards the body of evidence on mEHT usefulness in therapy of PDAC patients, the article needs careful rewriting and better organization of results and figures: findings need to be presented in a clearer way.  The authors should also provide some other outcome measures for assessing mEHT efficiency (e.g. progression-free survival, tumor response).

We are grateful for your kind suggestions and criticism, which helped to make our manuscript clearer and more focused.

  1. The text needs careful editing, it contains sentences which are difficult to understand: e.g lines 50-52, 150-151, 246-247, 256-257.

Thank you, sentences had been revised.

  1. What was the staging of patients?

Pancreatic cancer patients not eligible for surgery having stage III and IV cancer were included in the study. Number of stage III and IV patients was 34 and 44, respectively. Methods had been updated.

  1. What are the mechanisms behinds selectivity of mEHT toward cancer cells? – please add into Introduction

Thank you for the suggestion. Introduction had been extended with the following. “Without going into details, mEHT exploits the biophysical differences of cancer cells that their energy absorption on the membrane rafts is different than those of healthy hosts, furthermore, damage associated molecular patterns (DAMPS) will occur and all of these eventually leads to programmed or immunogenic tumor cell death [25,26]”

  1. Please provide reference in the Introduction to “mEHT …acts as monotherapy” (line87)

Thank you for bringing our attention to this small inaccuracy. It has been found in preclinical studies that mEHT monotherapy has tumoricidal effects that may can potentially as a standalone treatment. Furthermore, a few case-studies has been also published, where only mEHT treatment was used. The sentence had been revised: “The mEHT method re-sensitizes the refractory cases, can act as monotherapy as preclinical results and case-studies suggested [25,27-29], and increases the survival time and QoL in parallel”.

  1. The results section is not presented in a clear way, it would help to summarize all the results in a table where the medians of overall survival (and other measures) would be given for the whole groups and subgroups (patients with metastasis, ascites etc) as well. Please, summarize also the influence of all the other factors you assessed (e.g. location of tumor, No. of mEHT sessions etc) on the overall survival and mEHT result. In the present form, it is difficult to follow all the findings.

Thank you for your kind suggestion, Results had been reorganized, additional tables and figures had been introduced, due to lack of place some of these were moved to Supplementary materials. Figures previously included in the Appendix had been moved to Supplementary materials as well.

  1. The graphs should be grouped into figure panels in the order in which they are presented step-by-step in the Results text.

Thank you for your kind suggestion, Results, including figures had been completely reorganized.

  1. Can you provide other measures for assessing the impact of mEHT? For example: data on partial response, stable disease, progression free disease or time to progression?

Progression-free survival and response data had been added to Results.

  1. Fig A1 and the relevant text (lines 173-179): You report “that patients with metastatic disease benefited most from mEHT” – could you please explain this statement? The shift in OS is about 4 months in the metastasis group, in case of patients without ascites it is about 11 months when mEHT was administered. Please explain what HR of 2.11 and 1.6019 (line 178) refer to. What was the portion of patients having ascites in the groups with and without metastasis? Could it affect the results?

Results had been completely rewritten. Data presentation on metastatic disease was corrected. The connection between ascites and metastases was also further discussed. Please refer to the updated Results and Discussion parts of our manuscript.

  1. Fig A2: there are no dotted lines and colored intervals. In the figure, A is Control group, B is mEHT group, in the figure legend A is “with” and B is “without” mEHT

Thank you, the issues had been fixed.

  1. Conclusion at the end of discussion (line 263-265) and sentence (270-271 “best results can be achieved in metastatic cases”) should be rephrased

Conclusions had been rephrased, updates had been made in line with the additional results from progression-free survivals.

Yours sincerely,

                                            Prof. Dr. Magdolna Dank

                                             Semmelweis University

Reviewer 3 Report

The authors performed a retrospective single-center case-control study to assess the additional benefit of Modulated electro-hyperthermic (mEHT) to conventional systemic therapy of PDAC. Their hypothesis of backed by sufficient results. The manuscript is well written. I believe this article would be appropriate for publication after a minor English check. 

Round 2

Reviewer 2 Report

The authors addressed all my comments; the manuscript has been improved. I recommend the article to be accepted after correcting minor errors detailed below:

Minor remarks:

- Please correct the following grammatical mistakes:

Lines 312-313 „The observations may suggest that metastatic PDAC patients may be one that group which can benefit the most from the mEHT treatment.

Lines 329-330 „The  above observation were also confirmed in the present study“

Line 337 - „Patients with no ascites formation responds

- Figure S1: The figure legend reads: „trend to better survival both with (A) and without (B) the presence of metastases“ Directly in the graphs: A: without metastasis and B: with metastasis – please make corrections in the figure legend text